# Developing an Instrument to Evaluate the Quality of Dementia Websites

**DOI:** 10.3390/healthcare11243163

**Published:** 2023-12-13

**Authors:** Yunshu Zhu, Ting Song, Zhenyu Zhang, Ping Yu

**Affiliations:** School of Computing and Information Technology, University of Wollongong, Wollongong, NSW 2522, Australia; yz978@uowmail.edu.au (Y.Z.); tsong@uow.edu.au (T.S.); zhenyu@uow.edu.au (Z.Z.)

**Keywords:** dementia, Alzheimer’s disease, website, information quality, user experience, website quality measurement

## Abstract

In today’s digital era, health information, especially for conditions like dementia, is crucial. This study aims to develop an instrument, demenTia wEbsite measSurement insTrument (TEST), through four steps: identifying existing instruments, determining criteria, selecting and revising measurement statements, and validating the instrument from March to August 2020. Five health informatics experts used the content validity ratio (CVR) test for validation. Thirteen evaluators compared Fleiss Kappa and intraclass correlation coefficient (ICC) values across four dementia websites using TEST and another tool, DISCERN. TEST consists of seven criteria and 25 measurement statements focusing on content quality (relevance, credibility, currency) and user experience (accessibility, interactivity, attractiveness, privacy). CVR = 1 confirmed all statements as essential. The TEST demonstrated stronger consistency and assessor agreement compared to DISCERN, measured by Fleiss Kappa and ICC. Overall, it is a robust tool for reliable and user-friendly dementia resources, ensuring health holistic information accessibility.

## 1. Introduction

The increasing global prevalence of dementia, characterized by a progressive decline in cognitive function and autonomy, has far-reaching consequences for individuals, families, caregivers, and society at large [1]. Alzheimer’s disease, which accounts for the majority of dementia cases, highlights its significance as a public health concern [2]. Forecasts suggest that by 2050, a staggering 131.5 million people worldwide will be impacted by this condition [3].

Considering the growing challenge of dementia, the demand for high-quality online resources has become more pressing than before [4]. Websites, due to their broad accessibility, have emerged as the primary platform for individuals seeking information and support [5]. However, amidst the vast online expanse, ensuring consistent quality is difficult [5,6]. The reliability of online information content significantly impacts users’ decision-making processes related to dementia [7,8,9]. For example, two-thirds of Australian individuals seeking information on dementia look for details about early detection and care management strategies online [10]. They also communicate with government agencies and healthcare organisations that publish the information through their websites [5,6]. Access to accurate online information can increase their understanding of dementia (e.g., early signs and symptoms, diagnoses, treatments, care and caregiving tips). This knowledge can promote a dementia-supportive environment, guide patients in adopting preventative lifestyles, shape healthcare decisions, and empower caregivers with the tools to provide better care and thus potentially determine their overall health outcomes [7,11].

The design of websites significantly influences how users of varied abilities interact with the information presented [12,13]. This extends beyond individuals directly impacted by dementia—encompassing not only those managing the condition, such as individuals with dementia, caregivers, families, and friends but also those seeking vital information about dementia online. For example, clear headers facilitate seamless transitions between webpages, streamlining information access. Furthermore, a robust website design accommodates diverse device resolutions and user needs, extending its benefits not solely to individuals with dementia but to their entire support network [12,14]. Users accessing these sites—whether through desktops, mobile phones, or smartwatches—should have language preference options for a more inclusive experience [15]. Additionally, considering that many individuals with dementia are older and may have visual or auditory impairments, online browsing can pose complexities [16]. Effective website designs address these challenges by incorporating features like adjustable font sizes, color contrasts, and provisions for reading aids and subtitles. While ensuring website accuracy and user-friendliness is crucial, there is a current gap in maintaining these standards. Urgently developing an evaluation tool would guarantee trustworthy sources and improve dementia website accessibility. This tool empowers both users and their supportive circles, promising a substantial enhancement in assistance and support for the entire dementia-affected community.

Previous research has developed various instruments, including questionnaires and checklists, to evaluate the quality of health websites. For example, the Health On the Net Foundation (HONcode) evaluates the reliability and credibility of health information [17], whilst Web Medica Acreditada [18] evaluates the content quality of health websites. However, the quality of certain instruments (e.g., Date, Author, References, Type, Sponsor (DARTS) [19] and WebMedQual Scale [20] remains debatable. Concerns arise due to ambiguous criteria definitions, leading to confusion among online users about evaluation outcomes [19]. Moreover, the time required to utilize some of these tools is prohibitive, making them less user-friendly [20].

Specifically focusing on dementia, two instruments have been developed—Guideline Recommendations identified in the Canadian Consensus Conference on Diagnosis and Treatment of Dementia (Guideline) [21] and Dementia Caregiving Evaluation Tool (DCET) [22]. Both instruments aim to evaluate whether dementia websites provide comprehensive information for people with dementia and their caregivers. For example, “Does the website explains the difference between normal aging, mild cognitive impairment, and dementia” from Guideline [21] and “Does the website have information about how to cope with washing and bathing for caregivers?” from DCET [22]. However, both instruments lack criteria to evaluate essential quality facets of information content (e.g., authorship and reliability of information). Furthermore, the Guideline focuses predominantly on the comprehensiveness of dementia information for those affected and their caregivers (e.g., diagnosis and treatment of dementia), neglecting vital quality features such as usability. In addition, DCET was developed two decades ago and now seems outdated given the progression in dementia understanding and technological advancements.

This review of existing instruments underscores a significant void in the field—a lack of tools specifically designed to evaluate dementia websites, encompassing both information quality and user experience. To address this gap, our study endeavors to consolidate the strengths of current assessment tools while introducing a novel instrument, demenTia wEbsites meaSurement insTrument (TEST), specifically tailored for evaluating dementia websites.

## 2. Materials and Methods

To develop the TEST, we employed a rigorous, four-step iterative process based on established protocols from March to August 2020 [23]. The steps are (1) existing instrument identification, (2) criteria determination, (3) measurement statement selection and revision, and (4) instrument validation. The instrument, designed to assess website quality, consists of specific criteria and corresponding measurement statements. Criteria are abstract rules for delineating essential features that affect website quality [20]. They embody the evaluator’s values regarding what is deemed crucial in determining the quality of a website. Measurement statements are observable attributes associated with a website’s content, design, or overall presentation. These statements provide insights into whether a website aligns with the set criteria.

### 2.1. Step 1: Existing Instrument Identification

A systematic review was conducted to identify the existing evaluation instruments for general health websites. The literature search spanned across four databases: PubMed, Scope, CINAHL Plus with full text, and Web of Science. The following terms, MeSH headings, and searching schema were used to identify peer-reviewed journal papers published in English from 2009 to 2019: (“health”) AND (“web” or “website” or “site” or “internet” or “online”) AND (“quality” or “design” or “evaluat*”) OR (“assessment” or “credibility” or “criteri*”) AND “information” (“*” referred to a wildcard). To ensure comprehensive coverage, our search strategy encompassed an examination of grey literature sources, including Google Scholar, and a review of forward tracked papers, as recommended by the most recent systematic review evaluating the quality of online dementia information intended for consumers (Appendix A).

The search results were imported into Endnote 9.0. After removing duplicates, a preliminary screening by title and abstract was conducted by one researcher (YZ). The remaining papers were read in full text and further screened by two researchers (YZ and TS) independently against the following criteria (Table 1). The consensus was reached by discussing with a third researcher (PY).

Data were managed in Endnote X9 and interpreted as Excel spreadsheets, recording the author(s) and year, country of origin, the focused health topic, instrument(s) used to evaluate the website, scoring system, and target user(s) of the instrument(s).

### 2.2. Step 2: Criteria Determination

The criteria describing the key quality attributes of dementia websites were extracted from the identified instruments. Each criterion’s definition was recorded in an Excel spreadsheet. The panel consisted of four health informatics specialists: two focused on dementia, and two specialized in digital health. These four experts were selected based on their specialized knowledge in dementia and digital health, essential for shaping criteria crucial to the TEST tool’s development. This diverse composition allowed for a comprehensive evaluation of the usability criteria essential for assessing the quality of dementia websites, aligned with two guiding principles: (1) feasibility, i.e., ensuring the criterion is easily understandable and usable by general health consumers, and (2) domain independence: i.e., enabling assessment or evaluation by individuals without specialized health training or expertise.

### 2.3. Step 3: Measurement Statement Selection and Revision

The measurement statements, initially extracted from other existing instruments, were tailored to the dementia website evaluation context. All these statements were extracted into an Excel spreadsheet, read, and analysed consecutively by one researcher (YZ), aligning them with the selection criteria (Table 2) and ensuring their relevance and clarity.

### 2.4. Step 4: Instrument Validation

The content validity ratio (CVR) was used to validate the measurement statements of the TEST. It refers to the consensus among “subject-matter evaluators” on how well each question measures the construct [24]. The CVR value was calculated using the following formula:(1)$$CVR=\frac{n_{e-(N/2)}}{N/2}$$

*n_e_* is the number of evaluators indicating an item as “essential”, and *N* is the number of evaluators.

The *CVR* value ranges from −1 (perfect disagreement) to +1 (perfect agreement), with a *CVR* value above zero indicating that more than half of evaluators agree that the measurement statement is essential. Five experts participated in the review process: two specializing in dementia, two in digital health, and one from the industry. Each expert evaluated the measurement criteria, definitions, and associated statements. The selection criteria focused on securing diverse perspectives and specialized knowledge, encompassing expertise in dementia, digital health, and industry experience for a comprehensive assessment. They rated on a three-point scale (not necessary, useful but not essential, and essential) how well each statement represented the intended criterion of the TEST.

Fleiss Kappa and intraclass correlation coefficient (ICC) were used to assess the consistency and agreement levels of scores generated for 13 evaluators [25]. These evaluators, all postgraduate IT students, were chosen for their adeptness in information technology and their ability to critically assess and evaluate online content. Fleiss Kappa, a widely accepted statistical measure for inter-rater agreement, is calculated by formula (2) [25]. Interpretation of reliability followed Landis and Koch criteria utilizing kappa (k), classifying agreement levels: values <0 indicate poor agreement, 0.01 to 0.20 imply slight agreement, 0.21 to 0.40 suggest fair agreement, 0.41 to 0.60 indicate moderate agreement, 0.61 to 0.80 reflect substantial agreement, and 0.81 to 1.00 signify almost perfect agreement. The ICC, on the other hand, is employed to assess the consistency levels among evaluators, ranging from 0 to 1, where values closer to 1 indicate higher consistency among evaluators.
(2)$$p_{j}=\frac{1}{Nn}\sum_{i=1}^{N}n_{ij},\;1=\sum_{j=1}^{k}p_{j}$$

*N* is the total number of subjects; n is the number of evaluations per subject; *k* is the number of evaluation scales.

The subject starts at *i* = 1, and the evaluation scale starts at *j* = 1. *n_ij_* is the number of evaluators who assess the *i*th subject to the *j*th evaluating scale.

We divided a total of 13 postgraduate IT students into two groups, i.e., Group A and Group B, each comprising six and seven individuals separately. Initially, Group A was asked to evaluate Websites I (www.dementia.org.au) (accessed on 16 September 2022) and II (https://www.alzheimersresearchuk.org) (accessed on 16 September 2022) using the TEST, while Group B evaluated Websites III (www.alzheimer.ca/en) (accessed on 16 September 2022) and IV (www.alzint.org) (accessed on 16 September 2022) using the DISCERN [26]. After two weeks, Group A evaluated Websites I and II using the DISCERN [26], and Group B evaluated Websites III and IV using the TEST. They also provided comments for further improvement. The resulting scores were then imported into IBM SPSS Statistics 26.0 for analysis [27].

## 3. Results

### 3.1. Step 1: Existing Instrument Identification

The primary search yielded 320 publications. After removing duplicates, 54 papers remained. Their titles and abstracts were manually screened against the inclusion and exclusion criteria, leading to 44 candidate papers. Of these, 12 were excluded as irrelevant after further analysing the full paper. Finally, 32 papers from 31 research groups were included in this study, including 2 articles that reported a continuous study by one group in three years (Figure 1) [28]. For the characteristics of the included papers, see Appendix B.

Sixteen instruments were identified that could be used to evaluate the quality of health websites. Of these, 12 (75%) were designed for measuring the general health websites, including (1) HONcode [17], (2) DISCERN [26], (3) Delphi Discussion Model [28], (4)

LIDA Instrument [29], (5) Quality Component Scoring System (QCSS) [30], (6) Date, Author, References, Type, Sponsor (DARTS) [31], (7) Minimum Standard of e-Health Code of Ethics 2.0 [32], (8) Coding Scheme [33], (9) JAMA Benchmarks [34], (10) Quality Checklist [35], (11) WebMedQual Scale [20], and (12) Health-Related Website Evaluation Form (HRWEF) [36]. Two instruments (12.5%) were designed for evaluating dementia websites, including the Guideline [21] and DCET [22]. One instrument (6.25%) was designed to evaluate abortion websites—Abortion Service Information Assessment Tool [37]—and one (6.25%) for concussion websites: A Custom-Developed Concussion Checklist (CONcheck) [38].

### 3.2. Step 2: Criteria Determination

The review of existing instruments identified in Step 1 showed that all reported characteristics of website quality could be evaluated in two dimensions: information quality and design quality. Ten criteria were extracted to evaluate these two dimensions of website quality—accuracy, completeness, relevance, credibility, currency, readability, accessibility, interactivity, attractiveness, and privacy.

Information quality can be evaluated by six criteria—accuracy, completeness, relevance, credibility, currency, and readability. In the panel discussion, three criteria—accuracy, integrity, and readability—were excluded from entering the new instrument for two reasons. First, assessing information accuracy and completeness requires expertise in dementia domain knowledge, which is not possible for most general health consumers. Second, readability can be difficult to evaluate due to differences in consumer information literacy. To achieve more objective results, assessing readability can be automated using online tools like readability-score.com (https://readable.com/, accessed on 16 September 2022). This platform utilizes multiple formulas and metrics to evaluate text readability. Employing such automated approaches ensures a standardized evaluation of content readability, surpassing the subjectivity of manual assessment methods [39]. Therefore, seven criteria were selected with the definition presented in Table 3. Each criterion has 1 to 28 measurement statements. In total, there are seven criteria with 118 measurements.

### 3.3. Step 3. Measurement Statement Selection and Revision

For the seven criteria selected in Step 2, we defined exclusion criteria for selecting the measurement statements for each criterion (Table 4).

The resulted TEST consists of seven criteria and 25 measurement statements. It evaluates the quality of dementia websites in two dimensions—information quality (three criteria: relevance, credibility, and currency) and user experience quality (four criteria: accessibility, interactivity, attractiveness, and privacy). Each criterion has one to seven measurement statement(s) (see Table 5). Thirteen measurement statements were assessed by a binary scale (Yes/No), scored as “No” for 1 point and “Yes” for 5 points. The remaining 12 were assessed by a five-point Likert scale, where each statement was anchored between 1 point (strongly disagree) and 5 points (strongly agree), with 3 designated to represent “neither agree nor disagree”, allowing respondents to express a neutral stance (Table 5).

### 3.4. Step 4: Instrument Validation

Content validation results showed that five evaluators assessed 25 measurement statements as essential, with a CVR score of 1. This score achieved the CVR content validation requirement (the minimum acceptable CVR score is 0.99) for five evaluators.

The Fleiss Kappa values of TEST ranged from 0.34 to 0.91, with an average score of 0.61 (Table 6). This indicates moderate to substantial agreement among the observers. Furthermore, the ICC value of TEST was 0.97, demonstrating perfect agreement and excellent consistency upon a high level of confidence interval (95%) from 0.95 to 0.98. Conversely, the DISCERN group shows the lowest kappa (0.34) and ICC values (0.80) with the lowest confidence intervals of 0.33 and 0.68.

## 4. Discussion

For the first time, we developed an instrument, TEST, after conducting comprehensive comparative research on measurement instruments used for dementia and general health websites. TEST allows users to evaluate both the information content and user experience of dementia websites. The results of the Kappa coefficient and ICC results demonstrate the high level of agreement and consistency among the evaluators in assessing the dementia website using the TEST, which suggests that the TEST is a reliable tool for assessing the quality of such websites, enhancing confidence in its effectiveness. The high ICC values obtained provide strong evidence of the suitability and applicability of the TEST. These findings suggest that the ratings provided by the evaluators are consistent and reliable within the study sample, indicating that the TEST can effectively assess the desired criteria within this specific population. Additionally, the higher values obtained with the TEST suggest that it outperforms DISCERN in evaluating the quality of dementia websites. This finding is noteworthy as it underscores the superiority and practicality of the TEST as an evaluation instrument.

In comparison with other quality evaluation instruments, the TEST has several unique characteristics. Firstly, it addresses the limitations of existing instruments used for evaluating dementia websites, such as The Guideline [21], which focuses solely on assessing dementia information content. In contrast, we extended the scope of assessment beyond website content alone. By incorporating additional factors such as usability, accessibility, and user experience, we aimed to provide a more comprehensive evaluation framework. Secondly, the TEST also improves upon the limitations of the DCET [22], which primarily evaluates Alzheimer’s disease information for informal caregivers on websites. Recognizing the need for a broader perspective, we enhanced the instrument to encompass the needs of general health consumers beyond Alzheimer’s disease. This expanded focus now encompasses families, friends, and all individuals seeking online dementia information. Our modified version considered the diverse challenges and requirements faced by general health consumers across various types of dementia. This enhancement allowed for a more inclusive and comprehensive assessment of dementia-related information [41]. Thirdly, many health studies have indeed utilized long questionnaire survey instruments to comprehensively cover the topic area [40,42]. However, it is important to consider the potential drawbacks associated with lengthy instruments. One significant concern is the occurrence of response fatigue, which can arise when participants become fatigued or disengaged due to the extensive length of the questionnaire [43]. This fatigue can ultimately diminish the usability of the questionnaire, particularly for consumers. For example, WebMedQual includes 98 measurement statements that take a consumer 20 to 90 min to rate a health website for online health consumers [18]. In contrast, the TEST is designed as a lightweight tool, consisting of only 25 measurement statements. This streamlined approach enables consumers to evaluate the quality of a dementia website within a more manageable timeframe of approximately 10–15 min. Consequently, the TEST is well suited for general consumers, as it minimizes response fatigue and maintains a high level of usability.

This study has several limitations. First, the candidate measurement instruments are selected from the literature included in the review, which may not cover all the instruments published. Second, each measurement statement in the TEST is equally weighted and thus only roughly reflects the quality of each criterion. Future research needs to develop a weighted scale scoring system based on the relative contribution of each measurement statement to more precisely evaluate the overall quality of each criterion. Third, despite multiple rounds of design and modification, the final TEST was only validated by 13 users. Further validation is required through questionnaire surveys in a larger population to further test its internal consistency, reliability, construct validity, and analysis of the structural relationships such as multiple regression analysis, factor analysis, and path analysis.

## 5. Conclusions

The TEST is designed for users to evaluate information content and user experience of dementia websites. The seven criteria with 25 measurement statements allow for a quick and manageable evaluation process. The selection of criteria and measurement statements was based on rigorous assessment of existing instruments validated by domain experts, ensuring comprehensive coverage. Content validation and inter-rater reliability and effectiveness of TEST were conducted to validate the TEST. Additionally, a comparative analysis involving 13 evaluators was performed to assess its performance in comparison to other instruments. Overall, the TEST provides a user-friendly and comprehensive tool for evaluating dementia websites.

## Figures and Tables

**Figure 1 healthcare-11-03163-f001:**
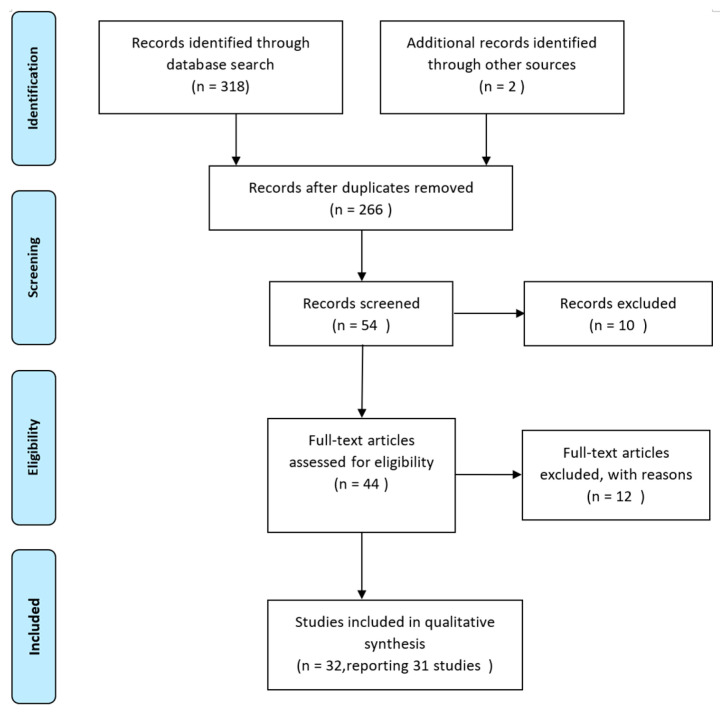
Literature search and screening process.

**Table 1 healthcare-11-03163-t001:** Selection criteria for existing instruments.

Inclusion criteria The study specifically evaluated health-related websites. The evaluation was focused on the quality of the websites. The evaluation method was described in detail.
Exclusion criteria The evaluation criteria were unclear. The evaluation was focused on social media platforms or software.

**Table 2 healthcare-11-03163-t002:** Selection criteria for the measurement statements.

Inclusion criteria The measurement statement only measures one item. The measurement statement can be answered by online consumers without dementia-specific knowledge. The statement measures the quality of dementia websites.
Exclusion criteria The measurement statement is used for guiding web designers to evaluate the accessibility of the website but may not be observable to online consumers. The measurement statement has no direct relationship with the quality of dementia websites. The measurement statement is vague or too general to allow objective assessment.

**Table 3 healthcare-11-03163-t003:** Selected criteria for evaluating dementia websites.

Criteria	Definition	References
Relevance	Relevance refers to the degree to which the information and hyperlinks are pertinent to the user’s needs.	[20,21,22]
Credibility	Credibility means the information is truthful or not biased. It is also referred to as trustworthiness. It is measured by three indicators: “Authorship”, “Attribution”, and “Disclosure”.	[17,20,29,32,34]
Currency	Currency refers to whether the content is up to date. The main indicators include the publication date and the time of the last update.	[19,20,32,40]
Accessibility	Accessibility refers to whether a website can be easily accessed and navigated around. The indicators include access by multiple languages; the provision of additional support; availability of search mechanisms; link to social media, relevant websites, and organisations; and providing access for consumers with disabilities.	[20,29,32]
Interactivity	Interactivity involves features that enable users to interact with the content, provide feedback, communicate with other users or the website’s author, and actively participate in various activities offered on the website. Essentially, it is about the responsiveness and ability of a system to engage users in two-way communication or participation rather than being a passive recipient of information. The indicators include whether the website provides opportunities to give feedback, whether users can exchange information with others, and whether the author can be contacted.	[20,29]
Attractiveness	Attractiveness refers to the look and feel of a site. The major indicators are site layout, the use of images, and the use of headings.	[20,29,32]
Privacy	Privacy refers to whether a website respects the privacy of personal data submitted by consumers. The privacy policy describes what information is collected, how it is used, and who can access it.	[20,22]

**Table 4 healthcare-11-03163-t004:** A representative sample of excluded measurement statements against the selection criteria.

Reason for Exclusion	Excluded Measurement Statements
The measurement statements guide web designers to evaluate the accessibility of the websites but may not be observable to the consumers.	“Does the site state expect response times for feedback?” “Does the site have any unnecessary links, layers, or clicks between documents or pages?” “Does the site provide instructions about how to disable cookies?” “Does the site present a policy statement or criteria for selecting links?” “Does the site clearly state that links have been reviewed?”
The measurement statements are about other diseases.	“Are links provided to regulate online and offline abortion services?”
The measurement statements require subjective evaluation.	“Are you in agreement with the entire website’s content?” “Is it easy to find the information you need?” “Is the content comprehensive within the given area?”
The measurement statements were vague or too general.	“Is the website organised logically?” “Is the website eye-pleasing?” “Is the page layout logical?”

**Table 5 healthcare-11-03163-t005:** The developed instrument demenTia wEbsite meaSurement insTrument (TEST) for general health consumers to evaluate the quality of dementia websites.

Criteria	Measurement Statement
Relevance	The website explores diverse aspects of dementia.
	☐ Yes ☐ No
	The website provides valuable insights for individuals living with dementia.
	☐ Strongly agree ☐ Agree ☐ Neither agree nor disagree ☐ Disagree ☐ Strongly Disagree
Credibility	The website cites references for the information presented.
	☐ Strongly agree ☐ Agree ☐ Neither agree nor disagree ☐ Disagree ☐ Strongly Disagree
	Any potential conflicts of interest arising from the website’s support are fully disclosed.
	☐ Strongly agree ☐ Agree ☐ Neither agree nor disagree ☐ Disagree ☐ Strongly Disagree
	The qualifications of the website owner(s) are clearly displayed.
	☐ Strongly agree ☐ Agree ☐ Neither agree nor disagree ☐ Disagree ☐ Strongly Disagree
	The author is recognized in the profession of health education or a related field.
	☐ Strongly agree ☐ Agree ☐ Neither agree nor disagree ☐ Disagree ☐ Strongly Disagree
	The website clearly states who is responsible for its content.
	☐ Yes ☐ No
Currency	The website displays the date of content creation.
	☐ Strongly agree ☐ Agree ☐ Neither agree nor disagree ☐ Disagree ☐ Strongly Disagree
	The website indicates the date when the content was last updated.
	☐ Strongly agree ☐ Agree ☐ Neither agree nor disagree ☐ Disagree ☐ Strongly Disagree
	The website provides recent events or advancements related to dementia.
	☐ Strongly agree ☐ Agree ☐ Neither agree nor disagree ☐ Disagree ☐ Strongly Disagree
Accessibility	The website features a user-friendly search mechanism, enabling visitors to find information efficiently.
	☐ Yes ☐ No
	The website offers content in the language preferred by the consumers.
	☐ Yes ☐ No
	The website is user-friendly for individuals with disabilities.
	☐ Yes ☐ No
	The website provides a link(s) to social media.
	☐ Yes ☐ No
	The website provides a link(s) to relevant external websites for further support and resources.
	☐ Yes ☐ No
	The website provides information about the related organization (health service or support organization).
	☐ Yes ☐ No
	The website connects users to other types of media (pamphlets, books, etc.) for additional information.
	☐ Strongly agree ☐ Agree ☐ Neither agree nor disagree ☐ Disagree ☐ Strongly Disagree
Interactivity	The website provides an opportunity for users to give feedback.
	☐ Yes ☐ No
	The website offers interactive features like discussion rooms or message boards for user engagement.
	☐ Yes ☐ No
	Author(s) can be contacted (by email, telephone or post).
	☐ Yes ☐ No
	The website layout and design are intuitive, enhancing my overall experience.
	☐ Strongly agree ☐ Agree ☐ Neither agree nor disagree ☐ Disagree ☐ Strongly Disagree
	I can easily navigate the website and find the information I am looking for.
	☐ Yes ☐ No
	I find the color scheme and visual elements engaging and pleasant.
	☐ Strongly agree ☐ Agree ☐ Neither agree nor disagree ☐ Disagree ☐ Strongly Disagree
	The website’s graphics enhance my understanding and engagement.
	☐ Strongly agree ☐ Agree ☐ Neither agree nor disagree ☐ Disagree ☐ Strongly Disagree
Privacy	The website states a privacy policy.
	☐ Yes ☐ No

**Table 6 healthcare-11-03163-t006:** Validation summary for measurement items.

Instrument	Kappa Value Mean (95% CI)	ICC Mean (95% CI)
TEST	0.61 (0.34–0.91)	0.97 (0.95–0.98)
DISCERN	0.34 (0.33–0.39)	0.80 (0.68–0.89)

## Data Availability

Data generated as part of this study are available from the corresponding author on reasonable request.

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
