# Peer review of "Developing an Instrument to Evaluate the Quality of Dementia Websites"

_healthcare, 2023, doi:10.3390/healthcare11243163_

Round 1
Reviewer 1 Report
Comments and Suggestions for Authors
Generally, the quality of the presentation and the scientific soundness make the manuscript clear, relevant for the field, and presented in a well-structured manner. However, substantial aspects need adequate attention to proceed with the publication (i.e., terms consistency). Below are my comments and suggestions, with the belief that they can contribute to making the value of this study more evident.
GENERAL
· The bibliography should be entirely revised following the Instructions for Authors and “MDPI Reference List and Citations Style Guide” (https://www.mdpi.com/authors/references)
· Reviewing the paper using the Strengthening the Reporting of Observational Studies in Epidemiology (STROBE) Statement: guidelines for reporting observational studies can be helpful.
INTRODUCTION
· Consider substituting the complex word "self-sufficiency" with "autonomy", that have in this context the same meaning, to improve the scientific soundness further.
· The following sentence seems to have some orthographic error: “There is an urgent need for high-quality online resources dedicated to dementia is more urgent than ever” (lines 37-38)
· From line 52 to line 70, the paragraph reports "anticipate" the study's problem statement, stating the importance of an appropriate and well-constructed website to be used by patients with dementia. Although, this population is fragile and, often, already in the early stages of the pathology, lacks "autonomy" (as stated by authors in the previous paragraphs). This problem highlights the need for formal and informal caregivers to support these patients' needs. So, in this context, why was the attention not focused on caregivers instead of patients?
METHODS
· The MEDLINE database is directly searchable from NLM as a subset of the PubMed® database. In other words, its references include the MEDLINE database. Moreover, PMC (PubMed Central) is a free archive for full-text biomedical and life sciences journal articles. Only some PMC content, such as book reviews, is not cited in PubMed. For these reasons, it can be said that the search was substantially done on PubMed, Scope, CINAHL Plus with full text, and Web of Science.
· One of the characteristics of systematic reviews is reproducibility and replicability. So, it is suggested to include the complete search strategy as a supplementary file.
· Moreover, considering that the author defined this search as a "systematic review", it is also necessary for a proper registration on Prospero. Was it done?
· Considering the availability of two instruments focused on dementia, it is unclear why it was decided to develop a new tool from instruments not focused on the pathology. Please specify.
METHODS
· Lines 157-159 states: "To evaluate inter-rater reliability, Fleiss Kappa was used across 13 evaluators, measuring the consistency among multiple evaluators assigning categorical ratings to a number of items”. Who are these “13 evaluators”? Health professionals? Informatics? Patients with dementia? Caregivers? Both? Please specify.
DISCUSSION
· While CVR and CVI are content validity indexes, Fleiss' kappa is a statistical measure for assessing the reliability of agreement between a fixed number of raters, not for measuring the instrument's reliability. Moreover, validity can be demonstrated by showing a clear relationship between the test and what it is meant to measure, assessing different types of validity: content validity, criterion-related validity, construct validity, or face validity. So, stating, "The TEST demonstrates acceptable content validity and reliability in the validation of the sample dementia websites” seems “speculative”. Please rephrase it both in discussion, conclusion and abstract.
In conclusion, some aspects have to be improved or refined. Therefore, this paper should be reviewed, and only after verification of the changes made can it be considered for publication.
Reviewer 2 Report
Comments and Suggestions for Authors
Thank you for the opportunity to review this manuscript on a current and relevant topic. Please find my suggestions and questions.
Manuscript
-Please, enter the date of the four-step process in the abstract and methods section.
-The objectives must be reviewed:
Lines 12-14: “This study aims to develop an instrument, demenTia wEbsite meaS urement insTrument (TEST) to evaluate the quality of dementia websites, including the information content and user experience design of websites.”
Lines 96-97: “To fill this gap, this study aims to amalgamate the strengths of the existing instruments and introduce a novel instrument for evaluating dementia websites - demenTia wEbsites meaSurement insTrument (TEST).”
Material and Methods
2.2. Step 2
Lines 132-133: “An expert panel of four health informatics experts deliberated on the usability of each criterion to evaluate the quality of dementia websites as per two principles…”
Where were the health informatics experts from? What were the criteria for choosing these four health informatics specialists?
2.4. Step 4
Lines 147-148: “Five evaluators from the health informatics domain scrutinized each measurement criterion, definition, and related measurement…”
Where were the evaluators from? What were the criteria for choosing these five evaluators?
Results
3.3. Step 3
-Line 218: I think the word “intractability” is wrong.
-Lines 218-219: “Each criterion has one to six measurement statement(s) (see Table 5).” However, “Accessibility” has seven measurement statements.
-Lines 220-221: “The rest 12 were assessed by a five-point Likert Scale. Each statement was anchored between 1 point (strongly disagree) and 5 points (strongly agree) (Table 5).” How about the option “neither agree nor disagree”? Can you describe the score? (0?)
3.4. Step 4
Lines 228-233: “Content validation results showed that five evaluators assessed 25 measurement statements as essential, with a CVR score of 1. This score achieved the CVR content validation requirement (the minimum acceptable CVR score is 0.99) for five evaluators (Appendix B). Five evaluators assessed 25 measurement statements as essential, with a CVR score of 1, above the minimum acceptable CVR score of 0.99 (see Appendix B).” The information is repeated.
References
The authors could include some references from the year 2023.
